# Induction of labour at 41 weeks or expectant management until 42 weeks: A systematic review and an individual participant data meta-analysis of randomised trials

Mårten Alkmark [1,2°‡*], Judit K. J. Keulen [3°‡], Joep C. Kortekaas [4], Christina Bergh [5,6], Jeroen van Dillen [4], Ruben G. Duijnhoven [3], Henrik Hagberg [1,2], Ben Willem Mol [7,8], Mattias Molin [9], Joris A. M. van der Post [3], Sissel Saltvedt [10], Anna-Karin Wikström [11], Ulla-Britt Wennerholm [1,2‡], Esteriek de Miranda [3°‡]

1 Centre of Perinatal Medicine & Health, Department of Obstetrics and Gynecology, Institute of Clinical Sciences, Sahlgrenska Academy, University of Gothenburg, Gothenburg, Sweden, 2 Department of Obstetrics, Sahlgrenska University Hospital, Region Vastra Gotaland, Gothenburg, Sweden, 3 Amsterdam UMC, University of Amsterdam, Department of Obstetrics and Gynaecology, Amsterdam Reproduction & Development Research Institute, Amsterdam, the Netherlands, 4 Department of Obstetrics and Gynaecology, Radboud University Medical Centre, Nijmegen, the Netherlands, 5 Department of Obstetrics and Gynecology, Institute of Clinical Science, Sahlgrenska Academy, Gothenburg University, Gothenburg, Sweden, 6 Department of Reproductive Medicine, Sahlgrenska University Hospital, Gothenburg, Sweden, 7 Department of Obstetrics and Gynaecology, Monash University, Monash Medical Centre, Clayton, Victoria, Australia, 8 Aberdeen Centre for Women's Health Research, University of Aberdeen, Aberdeen, United Kingdom, 9 Statistical Consulting Group, Gothenburg, Sweden, 10 Department of Women's and Children's Health, Karolinska Institute, Karolinska University Hospital, Stockholm, Sweden, 11 Department of Women's and Children's Health, Uppsala University, Uppsala, Sweden

☯ These authors contributed equally to this work.
‡ MA and JKJK share first authorship on this work. UBW and EdM are joint last author; both authors supervised the project.
* marten.alkmark@vgregion.se

**Data Availability Statement:** Data cannot be shared publicly directly because use of the data

## Abstract

### Background

The risk of perinatal death and severe neonatal morbidity increases gradually after 41 weeks of pregnancy. Several randomised controlled trials (RCTs) have assessed if induction of labour (IOL) in uncomplicated pregnancies at 41 weeks will improve perinatal outcomes. We performed an individual participant data meta-analysis (IPD-MA) on this subject.

### Methods and findings

We searched PubMed, Excerpta Medica dataBASE (Embase), The Cochrane Library, Cumulative Index of Nursing and Allied Health Literature (CINAHL), and PsycINFO on February 21, 2020 for RCTs comparing IOL at 41 weeks with expectant management until 42 weeks in women with uncomplicated pregnancies. Individual participant data (IPD) were sought from eligible RCTs. Primary outcome was a composite of severe adverse perinatal outcomes: mortality and severe neonatal morbidity. Additional outcomes included neonatal

from the individual trials was only approved by the ethical committee for use in this IPDA-MA. Approval for use for other purposes would be needed from the individual trials. Contact information for each study is given in supplement S6 Table.

**Funding:** UBW was supported by the Swedish state under the agreement between the Swedish government and the county councils, the ALF-agreement (ALFGBG-718721)(https://www.vr.se/english/mandates/clinical-research/clinical-research-within-the-alf-agreement.html). HH was supported by the Foundation of the Health and Medical care committee of the Region of Vastra Gotaland, Sweden (VGFOUREG854081) and UBW was supported by Hjalmar Svensson Foundation (https://www.stiftelsemedel.se/stiftelsen-handlanden-hjalmar-svenssons-forskningsfond/). The funders had no role in study design, data collection and analysis, decision to publish, or preparation of the manuscript.

**Competing interests:** I have read the journal's policy and the authors of this manuscript have the following competing interests: BWM is supported by a NHMRC Practitioner Fellowship (GNT1082548) BWM reports consultancy for ObsEva, Merck Merck KGaA and Guerbet

**Abbreviations:** ARR, absolute risk reduction; BMI, body mass index; CI, confidence interval; CINAHL, Cumulative Index of Nursing and Allied Health Literature; EEG, electroencephalography; Embase, Excerpta Medica database; EUROCAT, European Surveillance of Congenital Anomalies; HELLP, haemolysis, elevated liver enzymes, and a low platelet count; HTA, Health Technology Assessment; ICD-10, International Classification of Diseases 10th Revision; INDEX, INDuction of labour at 41 weeks versus a policy of EXpectant management until 42 weeks; IOL, induction of labour; IPD, individual participant data; IPD-MA, individual participant data meta-analysis; IQR, interquartile range; ITT, intention-to-treat; MA, meta-analysis; MAS, meconium aspiration syndrome; NICU, neonatal intensive care unit; NNT, number needed to treat; OR, odds ratio; PRISMA, Preferred Reporting Items for Systematic Reviews and Meta-Analyses; RCT, randomised controlled trial; RD, risk difference; RR, relative risk; SGA, small for gestational age; SWEPIS, SWEdish Post-term Induction Study.

admission, mode of delivery, perineal lacerations, and postpartum haemorrhage. Prespecified subgroup analyses were conducted for parity (nulliparous/multiparous), maternal age (<35/$\geq$35 years), and body mass index (BMI) (<30/$\geq$30). Aggregate data meta-analysis (MA) was performed to include data from RCTs for which IPD was not available.

From 89 full-text articles, we identified three eligible RCTs ($n$ = 5,161), and two contributed with IPD ($n$ = 4,561). Baseline characteristics were similar between the groups regarding age, parity, BMI, and higher level of education. IOL resulted overall in a decrease of severe adverse perinatal outcome (0.4% [10/2,281] versus 1.0% [23/2,280]; relative risk [RR] 0.43 [95% confidence interval [CI] 0.21 to 0.91], $p$-value 0.027, risk difference [RD] −57/10,000 [95% CI −106/10,000 to −8/10,000], $I^2$ 0%). The number needed to treat (NNT) was 175 (95% CI 94 to 1,267).

Perinatal deaths occurred in one (<0.1%) versus eight (0.4%) pregnancies (Peto odds ratio [OR] 0.21 [95% CI 0.06 to 0.78], $p$-value 0.019, RD −31/10,000, [95% CI −56/10,000 to −5/10,000], $I^2$ 0%, NNT 326, [95% CI 177 to 2,014]) and admission to a neonatal care unit $\geq$4 days occurred in 1.1% (24/2,280) versus 1.9% (46/2,273), (RR 0.52 [95% CI 0.32 to 0.85], $p$-value 0.009, RD −97/10,000 [95% CI −169/10,000 to −26/10,000], $I^2$ 0%, NNT 103 [95% CI 59 to 385]). There was no difference in the rate of cesarean delivery (10.5% versus 10.7%; RR 0.98, [95% CI 0.83 to 1.16], $p$-value 0.81) nor in other important perinatal, delivery, and maternal outcomes. MA on aggregate data showed similar results.

Prespecified subgroup analyses for the primary outcome showed a significant difference in the treatment effect ($p$ = 0.01 for interaction) for parity, but not for maternal age or BMI. The risk of severe adverse perinatal outcome was decreased for nulliparous women in the IOL group (0.3% [4/1,219] versus 1.6% [20/1,264]; RR 0.20 [95% CI 0.07 to 0.60], $p$-value 0.004, RD −127/10,000, [95% CI −204/10,000 to −50/10,000], $I^2$ 0%, NNT 79 [95% CI 49 to 201]) but not for multiparous women (0.6% [6/1,219] versus 0.3% [3/1,264]; RR 1.59 [95% CI 0.15 to 17.30], $p$-value 0.35, RD 27/10,000, [95% CI −29/10,000 to 84/10,000], $I^2$ 55%).

A limitation of this IPD-MA was the risk of overestimation of the effect on perinatal mortality due to early stopping of the largest included trial for safety reasons after the advice of the Data and Safety Monitoring Board. Furthermore, only two RCTs were eligible for the IPD-MA; thus, the possibility to assess severe adverse neonatal outcomes with few events was limited.

## Conclusions

In this study, we found that, overall, IOL at 41 weeks improved perinatal outcome compared with expectant management until 42 weeks without increasing the cesarean delivery rate. This benefit is shown only in nulliparous women, whereas for multiparous women, the incidence of mortality and morbidity was too low to demonstrate any effect. The magnitude of risk reduction of perinatal mortality remains uncertain. Women with pregnancies approaching 41 weeks should be informed on the risk differences according to parity so that they are able to make an informed choice for IOL at 41 weeks or expectant management until 42 weeks.

**Study Registration:** PROSPERO CRD42020163174

## Author summary

### Why was this study done?

- Timely induction of labour (IOL) aims to prevent adverse outcomes. In observational studies, although not usually in interventional studies, IOL has been associated with increased risks of emergency cesarean delivery, uterine hyperstimulation, and uterine rupture.

- The risk of stillbirth and several other serious perinatal and maternal complications increases as the pregnancy continues beyond term.

- According to a recent meta-analysis (MA) on aggregate data from randomised controlled trials (RCTs), perinatal mortality was lower after IOL at or beyond term compared with a policy of expectant management. However, the upper limit of gestational age for expectant management was not taken into account in this MA.

- The aim of this individual participant data meta-analysis (IPD-MA) was to compare the effect of a management strategy of IOL at 41 weeks versus expectant management until 42 weeks on important perinatal and maternal outcomes of women with low-risk singleton pregnancies as well as to identify subgroups of women that could benefit from IOL at 41 weeks.

### What did the researchers do and find?

- Three RCTs including a total of 5,161 women with low-risk singleton pregnancies comparing IOL at 41 gestational weeks with expectant management until 42 gestational weeks were identified.

- Data of two RCTs were available for inclusion in an IPD-MA with a total of 4,561 women.

- Overall, induction at 41 gestational weeks significantly reduced the composite outcome of perinatal mortality and severe neonatal morbidity, and perinatal mortality alone, without increasing the risk of cesarean delivery, operative vaginal delivery, perineal lacerations III and IV, or postpartum haemorrhage. However, the magnitude of the risk of perinatal mortality remains uncertain.

- A prespecified subgroup analysis showed that the risk of composite severe adverse perinatal outcome in the IOL group was significantly decreased for nulliparous, but not for multiparous women.

### What do these findings mean?

- Women with pregnancies approaching 41 gestational weeks should be informed of the benefits and risks of IOL at 41 weeks compared with expectant management until 42 weeks, with respect to the risk differences for nulliparous and multiparous women.

- Women can then make an informed choice regarding IOL at 41 weeks or awaiting spontaneous onset of labour until 42 weeks.

## Introduction

When to induce labour in overdue pregnancies has been under debate since many years. The risk of perinatal death and severe neonatal morbidity increases gradually after 41 weeks of pregnancy with a steeper increase after 42 weeks [1,2]. The proportion of women reaching 41 weeks varies in high-income countries between 5% and 25% [3]. In 2018, 22% of women in Sweden reached 41 weeks, and in the Netherlands, the rate was 16% [4,5].

Two randomised controlled trials (RCTs) recently evaluated the risk of adverse perinatal outcome after a policy of induction of labour (IOL) at 41 weeks as compared with expectant management until 42 weeks [6,7]. The first RCT concerns a non-inferiority trial among low-risk women in the Netherlands (INDuction of labour at 41 weeks versus a policy of EXpectant management until 42 weeks [INDEX]) comparing IOL at 41 weeks+0–1 days (41+0–1) with expectant management until 42 weeks+0 days (42+0), in which non-inferiority of expectant management was not proven [6]. The second RCT is a superiority trial from Sweden (SWEdish Post-term Induction Study [SWEPIS]) comparing induction at 41+0–2 with expectant management until 42+0–1 in low-risk women. It was stopped early because of safety reasons. The Data and Safety Monitoring Board recommended to stop the study owing to a higher perinatal mortality in the expectant management group (no perinatal mortality in the IOL group and six in the expectant management group). However, there was no significant difference in the primary outcome [7].

A Cochrane review from 2018 comparing birth outcomes after IOL or expectant management concluded that a policy of IOL at or beyond term is associated with fewer adverse perinatal outcomes and fewer cesarean deliveries compared with expectant management, though the absolute risk of perinatal death is small [8]. The conclusion is based on all included studies of the systematic review, but a majority of the included trials had expectant management groups that allows expectant management until far beyond 42 weeks, which could have influenced the overall outcomes. The authors suggested that women could be helped in deciding on IOL or expectant management by appropriate counselling (including information on absolute risks). Further exploration of risk profiles of women was recommended as well as individual participant data meta-analysis (IPD-MA), which could help elucidate the role of specific factors, such as parity, on outcomes of induction compared with expectant management. After results of the current study were already finalised, an update of this Cochrane review was published in which the main conclusion was similar as the previous version [9]. Although subgroup analysis was performed for parity, this was not done for the 41 to 42 weeks' comparison in this review. A recently published systematic review including two RCTs, two quasi-experimental trials, and three retrospective cohort studies compared IOL at 41+1 to 41+6 weeks with expectant management until 42+0 to 42+6 weeks. IOL at 41+1 to 41+6 weeks was found to be associated with an increased risk of cesarean delivery and pH <7.10, but it lacked power to estimate the risk of perinatal mortality [10]. However, the time frame in this systematic review allows comparison beyond 41 weeks with expectant management until 43 weeks, which is not comparable with our time frame. We chose an upper limit of 42 weeks because continuing pregnancy after 42 weeks is no longer regular policy since many years due to its association with increased perinatal mortality. This is reflected in many national and international guidelines [11–15]. Furthermore, nonexperimental studies without clear randomisation procedure are prone for selection bias. Also, we aimed to evaluate outcomes of intended management strategies, not actual start of labour, because women have to decide on a management strategy before they know if and when they will go into spontaneous onset of labour. For this reason, we only included RCTs in our IPD-MA.

Sample sizes of RCTs are typically insufficient to estimate the risk for rare outcomes like perinatal mortality and severe morbidity. Exploring potential subgroup effects of maternal age, parity, body mass index (BMI), or fetal sex is therefore impossible in individual trials. IPD-MA increases power and has the advantage to allow investigation of interactions between intervention and participant characteristics in the total RCT population as well as in subgroups [16].

The objective of the IPD-MA was to evaluate the effect of IOL at 41 weeks versus expectant management until 42 weeks on perinatal and maternal outcomes, with a focus on rare adverse perinatal outcomes. We also aimed to assess whether treatment effects differed in subgroups.

## Methods

The protocol for this IPD-MA was registered on PROSPERO (2020; CRD42020163174) and is available at https://www.crd.york.ac.uk/prospero/display_record.php?RecordID=163174. The IPD-MA is reported according to the Preferred Reporting Items for Systematic Reviews and Meta-Analyses (PRISMA)-IPD guidelines; see S1 Text PRISMA checklist [17].

Data from the INDEX trial were pseudo-anonymised. A code was generated for each participant, and the key to the personal data is stored in the archives of the participating midwifery practices and hospitals. Data in the SWEPIS trial were not anonymised. All data in the merged database were fully anonymised. A collaboration group from the SWEPIS and INDEX project teams conducted the study.

### Ethics statement

RCTs included in the IPD-MA had received country specific ethical approval for the study, and each participant gave informed written consent. Details can be found in the original manuscripts. Specific ancillary approval for the use of individual patient data for the purpose of this meta-analysis (MA) was given by the Medical Ethical Committee of Amsterdam UMC-AMC (Dnr W20_225#20.259, May 20, 2020) for reuse of data from the INDEX trial and from the SWEPIS trial by the Swedish Ethical Review Authority (Dnr 285–14, amendment 2019–04094, August 5, 2019).

### Specific objectives

The objectives of the IPD-MA were to apply IPD-MA methodology to assess the effect on perinatal and maternal outcomes, with a focus on rare adverse perinatal outcomes, after IOL at 41 weeks compared with expectant management until 42 weeks in women with low-risk singleton pregnancies and to identify possible subgroups that might benefit from IOL at 41 weeks. For this reason, the effect of the intervention was analysed for prespecified subgroups of participant characteristics: maternal age, parity, and BMI. In addition, a post hoc analysis on fetal sex was carried out.

### Eligibility criteria

RCTs were included if a strategy of IOL at 41+0–2 was compared with a strategy of expectant management with various regimes of fetal surveillance and induction at 42+0–1 in low-risk women with an uncomplicated singleton pregnancy. Perinatal mortality, neonatal, and maternal morbidity had to be reported. Only RCTs were eligible. Cluster-randomised RCTs and quasi-random design studies were not considered.

### Study identification: Information sources and search strategy

We performed a systematic literature search in PubMed, Excerpta Medica dataBASE (Embase), The Cochrane Library, Cumulative Index of Nursing and Allied Health Literature

(CINAHL), and PsycINFO (EBSCOhost Research Databases). Our search was a continuation of a systematic literature search made in the context of a Health Technology Assessment (HTA) report in 2012 exploring the same research question but with a wider inclusion criterion than in this paper [18]. UBW, HH, and CB authored this report. The searches comprised the time period from database inception to February 21, 2020. Reference lists of relevant articles were scrutinised for additional references. The detailed search strategy is presented in the Supporting information (S2 Text). We searched Clinicaltrials.gov and the World Health Organization's International Clinical Trials Registry Platform databases for ongoing and unpublished RCTs. We used no limitation for publication year or language.

### Study selection processes

Two HTA librarians carried out a first selection of eligible RCTs. Potential eligible RCTs were scrutinised independently by two teams (MA, HH, and UBW and JKJK, JCK, and EdM). Eligibility of RCTs was decided on at a consensus meeting.

### Data collection process and data items

Corresponding authors of eligible RCTs were contacted and invited to participate and provide data to the IPD-MA.

Prespecified variables and outcomes were discussed and defined by the authors. Study-level and participant-level data were scrutinised on the proportion of missing variables and data. The definitions of variables and outcomes were compared and harmonised. If a definition did not correspond, a new definition was agreed upon. In case of neonatal complications, e.g., meconium aspiration syndrome (MAS), intracranial bleeding, and neonatal infection/sepsis, we took advice from neonatologists and based the definition on the International Classification of Diseases 10th Revision (ICD-10) description [19].

IPD were collected on all randomised women. These included baseline data for descriptive purpose and analyses, date of randomisation, gestational age at randomisation, date of delivery, and data on primary and secondary outcomes. Data were checked for extreme or missing values and consistency with published data. The IPD was collected by one of the statisticians (MM), who managed the data and merged all data into one anonymised IPD-MA dataset. The IPD-MA dataset was stored in a secure database accessible only by the two statisticians (MM and RGD).

### IPD integrity

Sequence generation, data accuracy, data consistency and completeness, frequencies, and possible baseline imbalances of all outcomes used in this IPD analysis were checked by statisticians and authors of the included RCTs.

### Study quality including risk of bias assessment

We used the risk of bias tool developed by Cochrane to assess the risk of bias for each study [20]. Risk of bias was assessed independently by some authors (MA, HH, and UBW and JKJK, JCK, and EdM), and disagreement was resolved by discussion. Each study was evaluated for adequacy of randomisation (selection bias), blinding for participants and personnel and statistician responsible for analysis (performance bias), blinding of outcome assessment (detection bias), incomplete outcome data (attrition bias), selective reporting (reporting bias), and other bias (conflict of interest). Each risk of bias was rated as either low, unclear, moderate, or high for the RCT.

Two individual teams (MA, UBW, and HH and JKJK, JCK, and EdM, respectively) evaluated external validity, internal validity (risk of bias/study limitations), and precision for each study.

## Specification of outcomes and effect measures

The primary outcome "severe adverse perinatal outcome" was a composite of perinatal mortality and severe neonatal morbidity. Perinatal mortality was defined as stillbirth or neonatal mortality of live births with death between day 0 and 28 (deaths due to accidents were excluded). Severe neonatal morbidity was a composite of (1) five-minute Apgar score <4 (as an Apgar score <4 at five minutes is associated with an increased risk on long-term adverse neonatal outcome [21]); (2) hypoxic ischemic encephalopathy II-III (asphyxia/encephalopathy in need of therapeutic cooling); (3) intracranial haemorrhage (intracranial or intraventricular haemorrhage based on radiological findings with ultrasound of the brain, computed tomography, or magnetic resonance imaging); (4) neonatal convulsions (seizures with electroencephalography [EEG]/amplitude EEG confirmation, seizures without EEG/amplitude EEG confirmation, and silent seizures [EEG/diagnosis]); (5) MAS (respiratory distress after birth in the presence of meconium stained amniotic fluid with need of mechanical ventilation); (6) mechanical ventilation within the first 72 hours (with laryngeal tube and ventilator machine); and/or (7) obstetric brachial plexus injury.

Secondary perinatal outcomes consisted of all individual components of the composite outcome separately including stillbirth and neonatal mortality. Additional secondary outcomes were the composite outcome with five-minute Apgar <7 instead of Apgar <4, admission to neonatal care (medium care—excluding observation only for protocol—or intensive care unit), admission to neonatal care ≥4 days mimicking more intensified neonatal care for sick infants in need for longer treatment and/or more extensive observation (neonatal intensive care unit [NICU] admission as such could not be used as a variable due to different admission criteria in Sweden and the Netherlands), mean birthweight, small for gestational age (SGA) according to national birthweight curves (<10th percentile and <3rd percentile) [22,23], macrosomia (≥4,500 g), five-minute Apgar score <7, infection/sepsis (clinical suspected findings or proved positive blood culture and antibiotic treatment), meconium stained amniotic fluid, humerus fracture, and congenital anomalies (any congenital anomalies after excluding minor congenital anomalies according to the European Surveillance of Congenital Anomalies [EUROCAT]) [24].

Secondary maternal outcomes included interval from randomisation to delivery, gestational age at time of delivery, onset of labour (spontaneous or IOL), oxytocin during labour (for IOL and/or augmentation), pain treatment during vaginal delivery (epidural anaesthesia/spinal anaesthesia/opiates), mode of delivery (spontaneous vaginal delivery, assisted vaginal delivery, or cesarean delivery with indication for intervention), episiotomy, perineal lacerations III and IV, postpartum haemorrhage (>1,000 ml and >2,000 ml), fever during labour (≥38˚C), antibiotics during labour (prophylaxis or therapy), manual removal of placenta (with or without haemorrhage >1,000 ml), hypertensive disorders of pregnancy including eclampsia and HELLP syndrome (haemolysis, elevated liver enzymes, and a low platelet count), maternal deep vein thrombosis or pulmonary embolism, admission to intensive care unit, and maternal death up to 42 days after delivery (deaths due to accidents excluded).

Outcomes were analysed as relative risks (RRs) and risk differences (RDs; expressed per 10,000 patients). For major outcomes, the number needed to treat (NNT) was provided as well.

## Synthesis methods and data analysis

General baseline characteristics were presented as frequencies with percentages, means, and standard deviations or medians with interquartile ranges (IQRs).

## Individual participant data meta-analysis

One-step MA was done for those outcomes where IPD could be used. For dichotomous outcomes, RR and RD were estimated using generalised linear models with a log- or identity-link and binomial distribution, respectively. A categorical covariable coding for the study was used to permit for within-study associations. For all risk estimates, a 95% confidence interval (CI) and $p$-value were calculated.

In case of a zero-event outcome in one arm of one or both trials, Peto odds ratios (ORs) were calculated using a two-step approach [25]. In case of zero events in both arms (double zero events) in all but one RCT, no risk estimates or inferential statistics were calculated because double zero events will add zero weight to the IPD-MA. Continuity correction for sparse events was not used.

For continuous outcomes, the mean difference was estimated with 95% CI using a general linear model, also with a categorical covariable coding for study. Chi-squared test was used for non-ordered categorical variables.

All randomised women with outcome data were included in the final analyses. Outcomes were analysed on an intention-to-treat (ITT) basis according to the treatment allocated by randomisation comparing IOL at 41+0–2 to expectant management until 42+0–1.

Heterogeneity between trials was explored by calculating the $I^2$ and $p$-value estimates of variability. Values of $I^2 \geq 50\%$ were considered to indicate meaningful heterogeneity. Due to the low number of eligible RCTs, funnel plots for assessment of publication bias were not used. Application of meta-regression to explain heterogeneity was not possible for outcomes analysed by two-step analysis. NNT for benefit with 95% CI was calculated as the inverse of the absolute risk reduction (ARR): 1/ARR.

In order to assess whether the effect of the intervention differed by prespecified subgroups, analyses were conducted for parity (nulliparous and multiparous), maternal age ($<35$ years and $\geq 35$ years), and BMI ($<30$ and $\geq 30$). In addition, a post hoc subgroup analysis on fetal sex was performed. A test for multiplicative interaction between intervention and maternal characteristics was performed by means of an interaction term in the regression model to examine whether intervention effects differed between subgroups. An interaction with a $p$-value $<0.05$ was considered to indicate that the effect of intervention differed between subgroups. Subgroup analyses were only performed on the primary composite outcome and the selected secondary outcomes perinatal mortality and cesarean delivery. In case of significant interaction for any of these outcomes, additional analysis was performed on all outcomes for this subgroup.

## Meta-analysis on aggregate data

If IPD was not available, MA was performed on aggregate data from eligible trials. In this situation, RRs were calculated using a Mantel–Haenszel fixed effect model, or Peto ORs were calculated as appropriate.

Statistical analyses were performed using SAS 9.4 (SAS Institute, Cary, North Carolina, United States). Review Manager (RevMan, Computer program. Version5.3. Copenhagen: The Nordic Cochrane Centre, The Cochrane Collaboration, 2014) was used to conduct the MA on aggregate data.

## Results

### Study selection and IPD obtained

The updated literature search resulted in 1,111 articles after removal of duplicates. Another 20 RCTs were added from the former HTA report from 2012 [18]. All articles were screened on title and abstract. We assessed 89 full-text articles, of which three RCTs published between 2005 and 2019 were eligible for IPD-MA (n = 5,161 participants), the Gelisen and colleagues trial, the INDEX trial, and the SWEPIS trial (Fig 1) [6,7,26].

Eight RCTs were identified in the ongoing RCT search, of which one was relevant for our IPD-MA and is expected to run until September 2022 (ISRCTN 83219789, "The Finnish randomised controlled multicentre trial on optimal timing of labour induction in nulliparous women with post-term pregnancy"). Four RCTs were already published, and three were not relevant for our research question due to deviating intervention, comparison group, or gestational age.

The corresponding authors in each eligible RCT were contacted in order to participate in the IPD-MA. The corresponding author from the Gelisen and colleagues trial replied that they could not participate in the IPD-MA because the original database was not available anymore. We therefore conducted an MA on aggregate data available in all three RCTs for outcomes with similar definitions and relevance for this research question. In total, the three RCTs included 5,161 women, (n = 600 from the Gelisen and colleagues trial, n = 1,801 from the INDEX trial, and n = 2,760 from the SWEPIS trial): 2,581 women were assigned to IOL, and 2,580 to expectant management. Two of 3 RCTs contributed data for the IPD-MA (the INDEX and SWEPIS RCTs). Hence, the IPD-MA included 4,561 women: 2,281 women were assigned to IOL and 2,280 to expectant management.

### Study characteristics

The characteristics of the included RCTs are shown in Table 1.

All three trials included low-risk singleton pregnancies with fetus in cephalic position and had previous cesarean delivery or other major uterine surgery as an exclusion criterion. In the two trials included in the IPD-MA, fetal surveillance in the expectant management group was performed according to local protocol (Table 1).

In Table 2, the baseline characteristics of the population included in the IPD-MA are shown. Baseline characteristics were similar between the IOL and expectant management group. Heterogeneity between the two included RCTs was found for parity and educational level. In SWEPIS, 55.2% (762/1,381) of the IOL group were nulliparous versus 54.6% (753/1,379) in the expectant management group. In the INDEX trial, 50.8% (457/900) in the IOL group were nulliparous compared to 56.7% (511/907) in the expectant management group (S1 Table). In SWEPIS, 64.6% (789/1,275) of women in the IOL group and 62.8% (780/1,242) in the expectant management group had education on university or similar level. In the INDEX trial, the distribution was 31.8% (286/900) versus 35.7% (322/901) (S1 Table).

### IPD integrity

There were no important issues concerning outcome data identified in checking IPD in the two participating RCTs (S2 Table). A few secondary outcomes showed different proportions between the RCTs, e.g., admission to NICU and neonatal medium care, neonatal infection, and use of antibiotics during labour. For NICU admission, the indications for admittance were not comparable in both RCTs resulting in an imbalance for NICU admission. Therefore, a new variable was defined for both studies: "admission to neonatal care ≥4 days" as proxy for

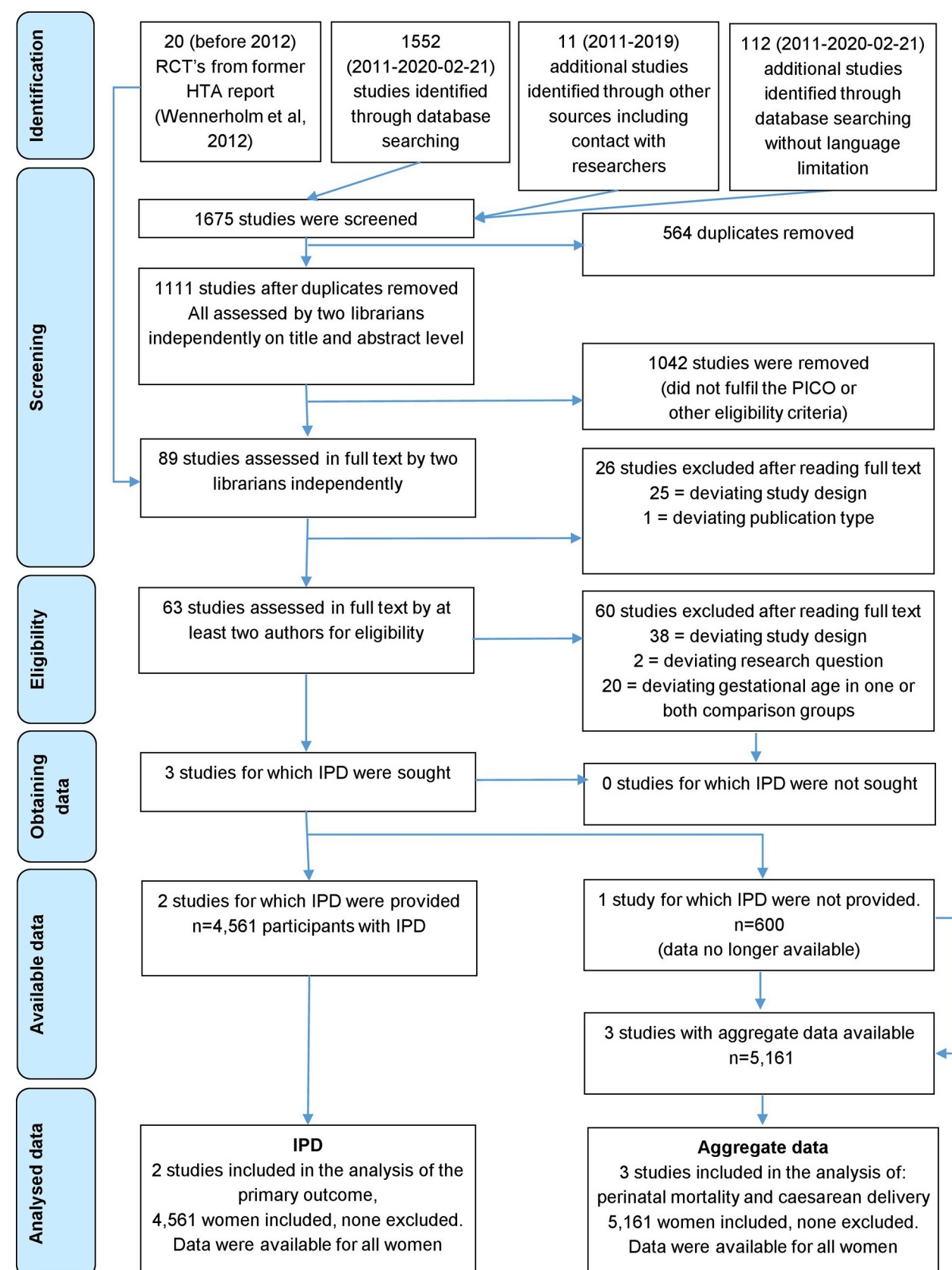

**Fig 1. PRISMA IPD flowchart of literature search.** HTA, Health Technology Assessment; IPD, individual participant data; PICO, patients, intervention, comparison, outcome; PRISMA, Preferred Reporting Items for Systematic Reviews and Meta-Analyses; RCT, randomised controlled trial.

infants in need for longer treatment and more extensive monitoring. Regarding the imbalance of neonatal infection, the INDEX trial included suspected infection in their outcome, and it was not possible to distinguish the true infections from the suspected ones. A plausible explanation for the discrepancy regarding maternal use of antibiotics during labour (significantly more in the SWEPIS trial) could be the difference in use of epidural anaesthesia, which is associated with fever and different policies regarding prophylactic treatment for group B streptococcus.

## Study quality including risk of bias assessment

Risk of bias within RCTs is presented in Table 3. The two RCTs included in the IPD-MA mostly had low risk of bias. In the Gelisen and colleagues trial, the risk of selection, detection,

**Table 1. Included RCTs in the aggregate MA and their characteristics.**

| Author, Year of publication, Country | Gelisen et al. 2005 Turkey | Keulen et al. 2019 The INDEX trial the Netherlands | Wennerholm et al. 2019 The SWEPIS trial Sweden |
|---|---|---|---|
| **Study design** | Single-centre superiority RCT | Multicentre, open-label, randomised controlled non-inferiority RCT | Multicentre, open-label, randomised controlled superiority RCT |
| **Participants** | 600 women with a Bishop score less than 5 (300 in IOL group and 300 in EM group) | 1,801 women regardless of Bishop score (900 in the IOL group and 901 in the EM group) | 2,760 women regardless of Bishop score (1,381 in the IOL group and 1,379 in the EM group). The RCT was planned to recruit 10,038 women, 5,019 in each arm. The RCT was stopped in advance due to safety reasons (6 perinatal deaths in EM group versus 0 in IOL group) |
| **Intervention** | IOL at 41+0–1 (three arms): (1) induction with 50-μg vaginal misoprostol (n = 100); (2) induction with transvaginal Foley balloon (n = 100); and (3) induction with infusion of oxytocin (n = 100) compared with EM until 42 +0. IOL method: 50-μg vaginal misoprostol. Fetal surveillance in the EM group: a nonstress cardiotocography test and amniotic fluid measurement twice weekly. In addition, a biophysical scoring on a single occasion three to five days after randomisation | IOL at 41+0–1 compared with EM until 42+0. IOL method in both groups according to local protocol, e.g., prostaglandin E1 (oral or vaginal), prostaglandin E2, Foley catheter or double-balloon catheter, or a combination of them and/or amniotomy. Fetal surveillance in the EM group was performed according to local protocols and could include cardiotocography and/or an ultrasound assessment of amniotic fluid | IOL at 41+0–2 compared with EM until 42+0–1. IOL method in both groups according to local management, e.g., prostaglandin E1 (oral or vaginal), prostaglandin E2, Foley catheter or double-balloon catheter, or a combination of them and/or amniotomy. Fetal surveillance in the EM group was performed according to local protocols and typically included antenatal visits with auscultation of the fetal heart rate |
| **Primary outcome** | Cesarean delivery rate, length of hospital stay, and neonatal outcomes (i.e., rate of macrosomia, incidence of meconium stained amniotic fluid, arterial cord blood pH <7.16, and rate of admission to NICU) | A composite of stillbirth, neonatal death until 28 days, Apgar<7 at five minutes and/or an arterial umbilical cord pH <7.05 and/or MAS and/or obstetric plexus brachialis injury and/or intracranial haemorrhage and/or NICU admission | A composite of stillbirth, neonatal death until 28 days, Apgar <7 at five minutes and/or an arterial umbilical cord pH <7.00 or metabolic acidosis (pH <7.05 and base deficit >12 mmol/L) and/or hypoxic ischaemic encephalopathy grades I–III and/or mechanical ventilation within 72 hours and/or intracranial haemorrhage, and/or convulsions and/or MAS and/or obstetric brachial plexus injury |
| **Follow-up** | Until discharge from the hospital | The neonates were followed until 28 days and the women 42 days postpartum regarding mortality. All other outcomes were followed until discharge from the hospital | The neonates were followed until 28 days and the women 42 days postpartum regarding mortality. All other outcomes were followed until discharge from the hospital |

EM, expectant management; INDEX, INDuction of labour at 41 weeks versus a policy of EXpectant management until 42 weeks; IOL, induction of labour; MA, meta-analysis; MAS, meconium aspiration syndrome; NICU, neonatal intensive care unit; RCT, randomised controlled trial; SWEPIS, SWEdish Post-term Induction Study.

**Table 2. Baseline characteristics of the population included in the individual patient data MA.**

| Variable | IOL group (*n* = 2,281) | EM group (*n* = 2,280) |
|---|---|---|
| Maternal age at randomisation (years) | *n* = 2,281 | *n* = 2,280 |
| Mean (standard deviation) | 31.0 (4.8) | 30.7 (4.6) |
| Age ≥35 | 479 (21.0) | 431 (18.9) |
| Parity (includes stillbirths and live births) | *n* = 2,281 | *n* = 2,280 |
| Nulliparous | 1,219 (53.4) | 1,264 (55.4) |
| Multiparous | 1,062 (46.6) | 1,016 (44.6) |
| BMI at first antenatal visit | *n* = 2,152* | *n* = 2,153* |
| Mean (standard deviation) | 24.8 (4.6) | 25.1 (4.8) |
| BMI ≥30 | 246 (11.4) | 301 (14.0) |
| Higher professional education/university | 1,075/2,121 (50.7)* | 1,102/2,143 (51.4)* |

Values are numbers (percentages) unless stated otherwise.

* Information on all participants was not available.

BMI, body mass index; EM, expectant management; IOL, induction of labour; MA, meta-analysis.

and reporting bias was unclear. In this RCT, published in 2004, sealed opaque envelopes were used for randomisation, the randomisation sequence process was not reported, and the study protocol was not published, which is all according to the standards at the time. All three RCTs had some risk of performance bias due to lack of blinding of participants and clinical personnel. In addition, the Gelisen and colleagues trial did not report if the assessment was blinded or not. In the INDEX trial, the statisticians who performed the analyses were blinded, but in the SWEPIS trial, they were not.

The Gelisen and colleagues trial had some or major problems regarding external validity and precision. The sample size was too small for detection of severe adverse perinatal outcomes. The INDEX trial had minor problems with external validity due to the low inclusion rate of eligible women (30%). The external validity was good to its own setting. Obstetric care in the Netherlands is divided into primary care (pregnancy and delivery of low-risk women supervised by a community midwife with delivery at home or in the hospital) and secondary care (pregnancy and delivery supervised by clinical midwives and obstetricians), which was reflected in the RCT. The RCT had no problems with precision for the primary outcome.

The SWEPIS trial had minor problems concerning external validity. There was a lower inclusion rate than expected (22% of eligible women), but compared to the background population, there were only small deviations including a higher rate of women with university

**Table 3. Risk of bias within individual RCTs.**

| Author | Gelisen et al. | INDEX trial | SWEPIS trial |
|---|---|---|---|
| **Selection bias** | Unclear | Low | Low |
| **Performance bias** | High* | Moderate* | High* |
| **Detection bias** | Unclear | Low | Low |
| **Attrition bias** | Low | Low | Low |
| **Reporting bias** | Unclear | Low | Low |
| **Conflict of interest bias** | Low | Low | Low |

* The lack of blinding in all RCTs is due to the nature of intervention, i.e., it is not possible to blind the participants and staff.

INDEX, INDuction of labour at 41 weeks versus a policy of EXpectant management until 42 weeks; RCT, randomised controlled trial; SWEPIS, SWEdish Post-term Induction Study.

education. In addition, the internal validity of the estimated risk for perinatal mortality can be affected by the early termination of the trial following advice from its Data and Safety Monitoring Board. The board recommended stopping for safety after an increased risk for perinatal mortality was observed at a planned interim analysis. Early termination of an RCT is associated with overestimating of the treatment effect, especially for events with low occurrence [27]. There were no or minor problems regarding study limitation for the other outcomes. Regarding precision, the study had lower power than planned for the composite outcome and perinatal mortality due to the early stopping.

## Primary outcome

In this IPD-MA, there was a significant difference in the primary composite outcome, in favour of the IOL group with 0.4% (10/2,281) compared to the expectant management group with 1.0% (23/2,280); (RR 0.43 [95% CI 0.21 to 0.91], *p*-value 0.027, RD −57/10,000 [95% CI −106/10,000 to −8/10,000], $I^2$ 0%) (Table 4). NNT was 175 (95% CI 94 to 1,267).

## Perinatal secondary outcomes

In Table 4, the perinatal outcomes are presented. Perinatal mortality was significantly lower in the IOL group with 0.04% (1/2,281) versus 0.35% (8/2,280) in the expectant management group (Peto OR 0.21 [95% CI 0.06 to 0.78], *p*-value 0.019, RD −31/10,000 [95% CI −56/10,000 to −5/10,000], $I^2$ 0%). NNT was 326 (95% CI 177 to 2,014).

The only perinatal death in the IOL group was a stillbirth that occurred one day after randomisation but before IOL. In the expectant management group, seven of the eight deaths were stillbirths and one baby died due to hypoxic ischemic encephalopathy. In postmortem examination of one of the stillbirths, a nonlethal cardiovascular malformation was found and two of the stillbirths were SGA (both between the 5th and 10th centiles). In the other six cases, no possible explanation was found.

Aggregate data MA on perinatal mortality of the three RCTs (*n* = 5,161 infants) showed similar results (S1 Fig). Since both included RCTs in the IPD-MA had a primary outcome with Apgar <7 instead of Apgar <4 at five minutes, we also calculated RR and RD for the primary outcome with Apgar <7. The primary outcome including Apgar <7 occurred in 1.5% (34/2,281) in the IOL group versus 2.3% (52/2,281) in the expectant management group (RR 0.65 [95% CI 0.43 to 1.00], *p*-value 0.051, RD −79/10,000 [95% CI −158/10,000 to 0], $I^2$ 13%).

Admittance to neonatal care unit for four days or longer was significantly lower in the IOL group compared with the expectant management group (1.1% [24/2,280] versus 2.0% [46/2,273], *p*-value 0.009, RR 0.52 [95% CI 0.32 to 0.85], RD −97/10,000 [95% CI −169/10,000 to −26/10,000], $I^2$ 0%). NNT was 103 (95% CI 59 to 385). There were fewer neonates with macrosomia (≥4,500 g) than in the expectant management group (3.9% [92/2,281] versus 6.7% [155/2,280], RR 0.59 [95% CI 0.46 to 0.76], *p*-value <0.001, RD −278/10,000 [95% CI −409/10,000 to −147/10,000], $I^2$ 0%). No significant differences were found in other perinatal secondary outcomes.

## Delivery outcomes

Table 5 summarises the delivery outcomes for the IPD-MA. The median gestational age in the IOL group was 288 (IQR 287; 289) days and 291 (IQR 289; 293) in the expectant management group, which corresponds with 41+1 versus 41+4 weeks, a difference of three days. In the IOL group, 79.8% (1,821/2,281) of the women were induced while the birth process started spontaneously in 19.9% (455/2,281 women). In the expectant management group, 30.4% (694/2,280) of the women were induced, while labour started spontaneously in 69.5% (1,584/2,280). In the

**Table 4. Perinatal outcomes in the population included in the IPD-MA.**

| Variable | IOL group (*n* = 2,281) | EM group (*n* = 2,280) | RR or Peto OR (95% CI) | *p*-value | Difference between groups RD per 10,000 or mean difference (95% CI) | Heterogeneity I² (%) | Heterogeneity *p*-value |
|---|---|---|---|---|---|---|---|
| **Primary outcome** | ***n* = 2,281** | ***n* = 2,280** | | | | | |
| Primary composite outcome* | 10 (0.4%) | 23 (1.0%) | 0.43 (0.21; 0.91)† | 0.027 | −57 (−106; −8) | 0 | 0.40 |
| **Subcomponents of primary composite outcome** | ***n* = 2,281** | ***n* = 2,280** | | | | | |
| Perinatal mortality‡ | 1 (0.0) | 8 (0.4) | 0.21 (0.06; 0.78)# | 0.019 | −31 (−56; −5) | 0 | 0.34 |
| Stillbirth | 1 (0.0) | 7 (0.3) | 0.22 (0.06; 0.89)# | 0.034 | −26 (−51; −2) | 0 | 0.36 |
| **Subcomponents of primary composite outcome** | ***n* = 2,280** | ***n* = 2,273** | | | | | |
| Neonatal mortality (live births with mortality <28 days) | 0 (0.0) | 1 (0.0) | NE | NE | NE | NE | NE |
| Apgar score <4 at 5 minutes of live births | 3 (0.1) | 4 (0.2) | 0.75 (0.17; 3.30)# | 0.70 | −4 (−27; 18) | 74 | 0.05 |
| HIE II-III | 2 (0.1) | 3 (0.1) | NE | NE | NE | NE | NE |
| Intracranial haemorrhage | 1 (0.0) | 2 (0.1) | NE | NE | NE | NE | NE |
| Neonatal convulsions | 1 (0.0) | 3 (0.1) | NE | NE | NE | NE | NE |
| MAS | 2 (0.1) | 5 (0.2) | 0.42 (0.10; 1.86)# | 0.25 | −13 (−36; 10) | 0 | 1.00 |
| Mechanical ventilation with tracheal intubation within the first 72 hours | 4 (0.2) | 9 (0.4) | 0.44 (0.14; 1.44)† | 0.18 | −22 (−53; 9) | 0 | 0.51 |
| Obstetric brachial plexus injury | 4 (0.2) | 1 (0.0) | NE | NE | NE | NE | NE |
| **Additional secondary neonatal outcome** | ***n* = 2,280** | ***n* = 2,273** | | | | | |
| Composite outcome with Apgar<7 at 5 minutes instead of <4 | 34/2,281 (1.5) | 52/2,280 (2.3) | 0.65 (0.43; 1.00)† | 0.051 | −79 (−158; −0) | 13 | 0.28 |
| Admittance to a neonatal care unit§ | 79 (3.5) | 109 (4.8) | 0.72 (0.54; 0.96)† | 0.024 | −133 (−249; −18) | 0 | 0.38 |
| Admission to a neonatal care unit ≥4 days | 24 (1.1) | 46 (1.9) | 0.52 (0.32; 0.85)† | 0.009 | −97 (−163; −26) | 0 | 0.35 |
| Neonatal infection or sepsis¶ | 49 (2.1) | 59 (2.6) | 0.83 (0.57; 1.21) | 0.33 | −44 (−132; 44) | 52 | 0.15 |
| Apgar score <7 at 5 minutes of live births | 29 (1.3) | 39 (1.7) | 0.74 (0.46; 1.19) | 0.22 | −44 (−115; 26) | 66 | 0.09 |
| Humerus fracture | 0/2,281 (0.0) | 1/2,280 (0.0) | NE | NE | NE | NE | NE |
| Birth weight (g) | ***n* = 2,281** | ***n* = 2,280** | | | | | |
| Mean (standard deviation) | 3,764 (417) | 3,823 (439) | | <0.001 | −58.6 (−83.5; −33.8) | | |
| Macrosomia (≥4,500 g) | 92 (3.9) | 155 (6.7) | 0.59 (0.46; 0.76)† | <0.001 | −278 (−409; −147) | 0 | 0.97 |
| SGA‖ | | | | | | | |
| <3rd percentile | 37 (1.6) | 45 (2.0) | 0.82 (0.54; 1.27) | 0.38 | −35 (−112; 42) | 83 | 0.01 |
| <10th percentile | 169 (7.4) | 188 (8.2) | 0.90 (0.74; 1.10)† | 0.29 | −84 (−239; 72) | 0 | 0.88 |
| Congenital anomaly** | 30 (1.3) | 36 (1.6) | 0.83 (0.52; 1.35)† | 0.46 | −26 (−96; 43) | 0 | 0.96 |
| Boy | 1,228 (53.8) | 1,194 (52.4) | 1.03 (0.97; 1.09)† | 0.31 | 146 (−143; 435) | 0 | 0.87 |

Values are numbers (percentages) unless stated otherwise. RR is adjusted for RCT. *p*-value corresponds to the method used to calculate the RR/OR.

* Including perinatal mortality, Apgar<4 at 5 minutes, HIE II-III, intracranial haemorrhage, neonatal convulsions, MAS, obstetric brachial plexus injury, and mechanical ventilation within 72 hours.

† Adjusted RR.

‡ Stillbirth and neonatal mortality (live births with mortality <28 days).

# Peto OR.

§ Neonates admitted only for routine observation excluded.

¶ In the INDEX trial, neonates with suspected infection are included.

‖ According to national gestational and sex-specific references [22,23].

** Minor birth anomalies according to EUROCAT excluded [24].

CI, confidence interval; EM, expectant management; EUROCAT, European Surveillance of Congenital Anomalies; HIE, hypoxic ischemic encephalopathy; INDEX, INDuction of labour at 41 weeks versus a policy of EXpectant management until 42 weeks; IOL, induction of labour; IPD-MA, individual participant data meta-analysis; MAS, meconium aspiration syndrome; NE, not estimated due to 0 events in both arms in 1 or both trials; OR, odds ratio; RCT, randomised controlled trial; RD, risk difference; RR, relative risk; SGA, small for gestational age.

**Table 5. Delivery outcomes in the population included in the IPD-MA.**

| Variable | IOL group (n = 2,281) | EM group (n = 2,280) | RR (95% CI) | p-value | Difference between groups RD per 10,000 or mean difference (95% CI) | Heterogeneity I²(%) | p-value |
|---|---|---|---|---|---|---|---|
| **Gestational age at delivery (days)** | **n = 2,281** | **n = 2,280** | | | | | |
| Median (IQR) | 288 (287; 289) | 291 (289; 293) | | <0.001 | | | |
| **Time from randomisation to delivery (days)** | **n = 2,281** | **n = 2,280** | | | | | |
| Mean (standard deviation) | 1.88 (1.49) | 4.47 (2.81) | | <0.001 | −2.59 (−2.72; −2.46) | | |
| **Onset of delivery** | **n = 2,281** | **n = 2,280** | | <0.001 | | | |
| Spontaneous | 455 (19.9) | 1,584 (69.5) | | | | | |
| Induction | 1,821 (79.8) | 694 (30.4) | | | | | |
| Scheduled cesarean delivery | 5 (0.2) | 2 (0.1) | | | | | |
| Meconium stained amniotic fluid | 380/2,138 (17.8) | 525/2,028 (25.9) | 0.68 (0.61; 0.77) | <0.001 | −821 (−1,070; −572) | 0 | 0.52 |
| Use of oxytocin* | 1,440/2,281 (63.1) | 1,077/2,280 (47.2) | 1.33 (1.26; 1.40) | <0.001 | 1,589 (1,305; 1,872) | 89 | 0.002 |
| **Mode of delivery** | **n = 2,281** | **n = 2,280** | | | | | |
| Spontaneous vaginal delivery | 1,860 (81.5) | 1,836 (80.5) | 1.01 (0.98; 1.04) | 0.41 | 101(−126; 328) | 0 | 0.65 |
| Cesarean delivery | 240 (10.5) | 245 (10.7) | 0.98 (0.83; 1.16) | 0.81 | −22 (−201; 157) | 0 | 0.83 |
| Operative vaginal delivery | 181 (7.9) | 199 (8.7) | 0.91 (0.75; 1.10) | 0.33 | −79 (−239; 81) | 0 | 0.56 |
| **Indication for cesarean delivery** | **n = 240** | **n = 245** | | 0.34¶ | | | |
| Failure to progress‡ | 120 (50.0) | 122 (49.8) | | | | | |
| Suspected fetal distress | 65 (27.0) | 57 (23.3) | | | | | |
| Suspected fetal distress and failure to progress | 17 (7.1) | 21 (8.6) | | | | | |
| Failed operative vaginal delivery | 13 (5.4) | 24 (9.8) | | | | | |
| Other§ | 25 (10.4) | 21 (8.6) | | | | | |
| **Indication for operative vaginal delivery** | **n = 181** | **n = 199** | | 0.24¶ | | | |
| Failure to progress| | 89 (49.2) | 98 (49.2) | | | | | |
| Fetal distress | 76 (42.0) | 71 (35.7) | | | | | |
| Fetal distress and failure to progress | 15 (8.3) | 29 (14.6) | | | | | |
| Maternal complication | 1 (0.6) | 1 (0.5) | | | | | |

Values are numbers (percentages) unless stated otherwise. RR is adjusted for trial.

* Both induction and/or labour augmentation.

‡ Including failed induction.

§ Including scheduled due to, e.g., undetected breech or transverse presentation/maternal indication.

¶ Chi-squared test.

| Including maternal distress.

CI, confidence interval; EM, expectant management; IOL, induction of labour; IPD-MA, individual participant data meta-analysis; IQR, interquartile range; RD, risk difference; RR, relative risk.

IOL group, 0.2% of the women (5/2,280) versus 0.1% (2/2,281) in the expectant management group had a scheduled cesarean delivery.

In addition, aggregate data MA of the three RCTs (n = 5,161 women) showed no difference in the frequency of cesarean delivery (S2 Fig).

In the IOL group, the presence of meconium-stained amniotic fluid was lower than in the expectant management group (17.8% [380/2,138] versus 25.9%, [525/2,028] RR 0.68 [95% CI 0.61 to 0.77], p-value < 0.001, RD −821/10,000 [95% CI −1,070/10,000 to −572/10,000], I² 0%). The rate of MAS in the IOL group was 0.1% (2/2,280) and 0.2% (5/2,273) in the expectant management group, RR 0.42 (0.10 to 1.86), p-value 0.25, RD −13/10,000 (−36/10,000 to 10/10,000), I² 0%. The use of oxytocin was higher in the IOL group compared to the expectant management group (63.1% [1,440/2,281] versus 47.2% [1,077/2,280], RR 1.33 [95% CI 1.26

**Table 6. Maternal outcomes in the population included in the IPD-MA.**

| Variable | IOL group (n = 2,281) | EM group (n = 2,280) | RR (95% CI) | p-value | RD per 10,000 (95% CI) | Heterogeneity | |
|---|---|---|---|---|---|---|---|
| | | | | | | I² (%) | p-value |
| Pain treatment (Use of epidural/spinal/opiates)* | 1,153 (50.5) | 1,058 (46.4) | 1.09 (1.03; 1.16) | 0.005 | 414 (125; 703) | 0 | 0.85 |
| Use of epidural anaesthesia | 998 (43.8) | 906 (39.7) | 1.10 (1.03; 1.17) | 0.006 | 400 (122; 678) | 0 | 0.50 |
| Use of opiates | 184 (8.1) | 173 (7.6) | NE | NE | NE | NE | NE |
| Fever during labour | 177 (7.8) | 151 (6.6) | 1.17 (0.95; 1.44) | 0.14 | 114 (−96; 195) | 0 | 0.66 |
| Antibiotics during labour | 296 (13.0) | 297 (13.0) | 0.99 (0.85; 1.15) | 0.88 | −6 (−197; 185) | 62 | 0.11 |
| Therapy | 128 (5.6) | 104 (4.6) | 1.23 (0.96; 1.58) | 0.11 | 105 (−22; 232) | 0 | 0.62 |
| Prophylaxis | 168 (7.4) | 193 (8.5) | 0.87 (0.71; 1.05) | 0.15 | −111 (−264; 43) | 30 | 0.23 |
| Episiotomy‡ | 328 (14.4) | 335 (14.7) | 0.97 (0.85; 1.11) | 0.71 | −30 (−226; 166) | 0 | 0.53 |
| Perineal lacerations III and IV§ | 68 (3.0) | 81 (3.6) | 0.84 (0.61; 1.15) | 0.28 | −57 (−160; 46) | 0 | 0.71 |
| Postpartum haemorrhage (>1,000 ml)¶ | 208 (9.1) | 204 (9.0) | 0.98 (0.82; 1.17) | 0.86 | 17 (−149; 184) | 70 | 0.067 |
| Postpartum haemorrhage (>2,000 ml)¶ | 42 (1.8) | 34 (1.5) | 1.23 (0.79; 1.93) | 0.36 | 35 (−39; 109) | 0 | 0.56 |
| Retained placenta (all) | 93 (4.1) | 90 (3.9) | 1.03 (0.78; 1.37) | 0.82 | 13 (−101; 127) | 10 | 0.29 |
| Retained placenta with haemorrhage (>1,000 ml) | 62 (2.7) | 61 (2.7) | 1.02 (0.72; 1.44) | 0.93 | 4 (−90; 98) | 0 | 0.91 |
| Retained placenta with haemorrhage (≤1,000 ml) | 31 (1.4) | 29 (1.3) | 1.07 (0.65; 1.77) | 0.80 | 9 (−57; 75) | 63 | 0.10 |
| Hypertensive disorders\| | 26 (1.1) | 66 (2.9) | 0.39 (0.25; 0.61) | <0.001 | −176 (−257; −94) | 0 | 0.39 |
| Maternal venous thromboembolism | 0 (0.0) | 1 (0.0) | NE | NE | NE | NE | NE |
| Maternal admission to intensive care unit | 5 (0.2) | 2 (0.1) | 2.50 (0.49; 12.86) | 0.27 | 13 (−10; 36) | 0 | NE |
| Maternal death | 0 (0.0) | 0 (0.0) | NE | NE | NE | NE | NE |

Values are numbers (percentages) unless stated otherwise. RR is adjusted for RCT.

* In the INDEX trial, a combination of epidural and opiates was possible.

‡ With and without perineal lacerations III and IV.

§ With and without episiotomy.

¶ Based on measured blood loss and not ICD-10 codes reported.

| Hypertensive disorders of pregnancy including eclampsia and HELLP.

CI, confidence interval; EM, expectant management; HELLP, haemolysis, elevated liver enzymes, and a low platelet count; ICD-10, International Classification of Diseases 10th Revision; INDEX, INDuction of labour at 41 weeks versus a policy of EXpectant management until 42 weeks; IOL, induction of labour; IPD-MA, individual participant data meta-analysis; NE, not estimated due to 0 events in both arms in 1 or both trials; RD, risk difference; RCT, randomised controlled trial; RR, relative risk.

to 1.40], p-value <0.001, RD 1,589/10,000 [95% CI 1,305/10,000 to 1,872/10,000], I² 89%). The use of oxytocin was overall higher in SWEPIS, 65.7% versus 52.4% compared with the INDEX trial 26.6% versus 10.9% (S3 Table).

There were 10.5% (240/2,281) cesarean deliveries in the IOL group versus 10.7% (245/2,280) in the expectant management group (RR 0.98 [95% CI 0.83 to 1.16], p-value 0.81, RD −22/10,000 [95% CI −201/10,000 to 157/10,000], I² 0%). Uterine hyperstimulation was not registered as such, but there was no increase in cesarean delivery after IOL compared to expectant management, and there was no difference between the groups with respect to the indication for cesarean delivery.

## Maternal secondary outcomes

The maternal outcomes are presented in Table 6. Use of pain treatment (epidural/spinal or opiates) was significantly higher in the IOL group compared to the expectant management

group (50.5% [1,153/2,281] versus 46.4%, [1,058/2,280], RR 1.09 [95% CI 1.03 to 1.16], *p*-value 0.005, RD 414/10,000 [95% CI 125/10,000 to 703/10,000], $I^2$ 0%). Opiates during delivery were only used in the INDEX trial.

The occurrence of hypertensive disorders of pregnancy was lower in the IOL than in the expectant management group (1.1% [26/2,281] versus 2.9% [66/2,280], RR 0.39 [95% CI 0.25 to 0.61], *p*-value <0.001, RD −176/10,000 [95% CI −257/10,000 to −94/10,000], $I^2$ 0%, NNT 57 [95% CI 39 to 106]). There were no differences in severe morbidity, such as perineal lacerations III and IV, postpartum haemorrhage, and rare events such as venous thromboembolism.

## Subgroup analysis

The subgroup analysis on the primary composite outcome and cesarean delivery is presented in Fig 2 and in S5 Table. The prespecified analysis on the primary composite outcome showed a significant difference in the treatment effect according to parity (*p* = 0.01 for interaction). The risk of adverse composite perinatal outcome in the IOL group versus the expectant management group was significantly decreased for nulliparous women (0.3% [4/1,219] versus 1.6% [20/1,264], RR 0.20 [95% CI 0.07 to 0.60], *p*-value 0.004, RD −127/10,000 [95% CI −204/10,000 to −50/10,000], $I^2$ 0%, NNT 79 [49 to 201]), but not for multiparous women (0.6% [6/1,062] versus 0.3% [3/1,016], RR 1.93 [95% CI 0.48 to 7.72], *p*-value 0.35, RD 27 [95% CI −29/10,000 to 84/10,000], $I^2$ 55%).

There was no significant difference in the treatment effect on the primary composite outcome according to maternal age (<35 years and ≥35 years) and BMI (<30 and ≥30) (*p* = 0.45 and *p* = 0.62, respectively, for interaction).

The post hoc subgroup analysis on fetal sex had a *p*-value for interaction of 0.1 on the primary composite outcome. In boys, the composite outcome in the IOL group versus the expectant management group occurred in 0.4% (5/1,228) versus 1.5% (18/1,194), (RR 0.28 [95% CI 0.10 to 0.75], *p*-value 0.01, RD −110/10,000 [95% CI −187/10,000 to −33/10,000], $I^2$ 0%) and in girls in 0.5% (5/1,053) versus 0.5% (5/1,086), (RR 1.05 [95% CI 0.30 to 3.72], *p*-value 0.96, RD 2/10,000 [95% CI −56/10,000 to 59/10,000], $I^2$ 0%).

Because of the low perinatal mortality rate (0.2%, *n* = 9), no interaction analysis on mortality could be performed. Perinatal mortality according to subgroups is presented in S5 Table. In nulliparous women, perinatal mortality occurred in 0% (0/1,219) in the IOL group versus 0.9% (7/1,264) in the expectant management group. In multiparous women, the corresponding figures were 0.1% (1/1,062) versus 0.1% (1/1,016).

There was no significant difference in the treatment effect on cesarean delivery according to parity, maternal age, or BMI (*p*-value for interaction 0.88, 0.07, and 0.32, respectively) (Fig 2, S5 Table). The rate of cesarean delivery in nulliparous women was 18.0% (219/1,219) in the IOL group and 17.9% (226/1,264) in the expectant management group (RR 1.01 [95% CI 0.85 to 1.19], *p*-value 0.95). Corresponding rates for multiparous women were 2.0% (21/1,062) and 1.9% (19/1,016) (RR 1.05 [95% CI 0.57 to 1.95], *p*-value 0.87).

For other outcomes, there were no significant interaction effects for parity, except for use of oxytocin. This was used more frequently in nulliparous women, 76% in the IOL group versus 65% in the expectant management group, and corresponding rates for multiparous women were 49% versus 26% (*p*-value for interaction <0.001) (S5 Table).

## Discussion

### Summary of evidence

Overall, we found in this IPD-MA that IOL in women with an uncomplicated pregnancy at 41 weeks reduced the incidence of severe adverse perinatal outcome as compared with expectant

### A. Primary outcome: severe adverse perinatal outcome*

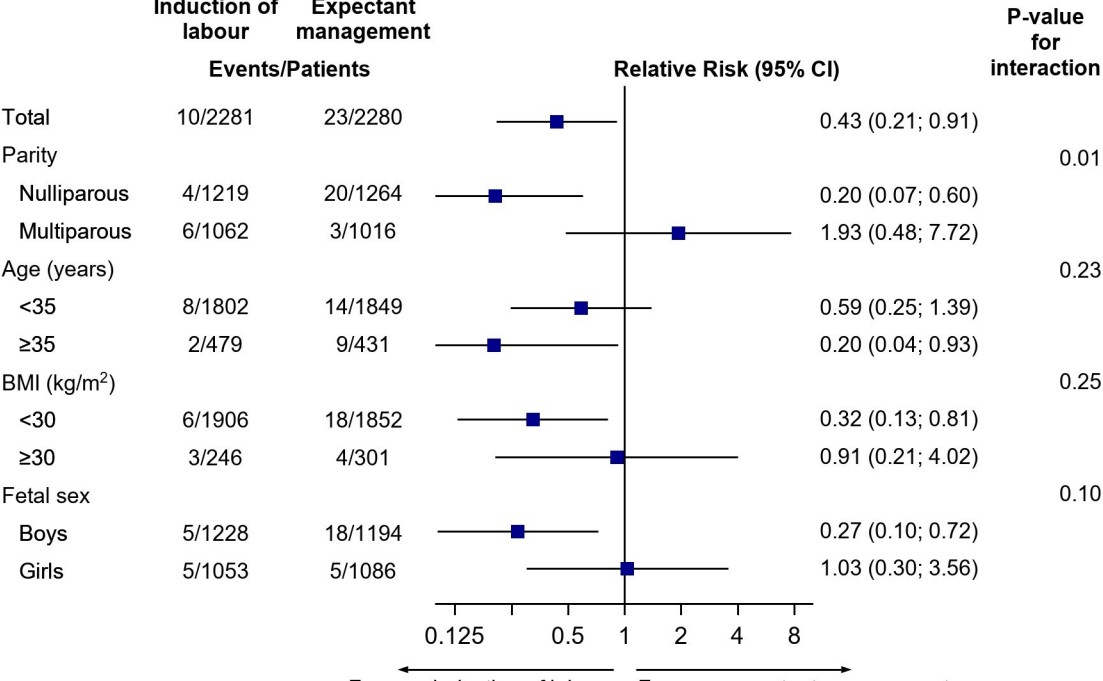

| | Induction of labour | Expectant management | Relative Risk (95% CI) | P-value for interaction |
|---|---|---|---|---|
| | Events/Patients | | | |
| Total | 10/2281 | 23/2280 | 0.43 (0.21; 0.91) | |
| Parity | | | | 0.01 |
| Nulliparous | 4/1219 | 20/1264 | 0.20 (0.07; 0.60) | |
| Multiparous | 6/1062 | 3/1016 | 1.93 (0.48; 7.72) | |
| Age (years) | | | | 0.23 |
| <35 | 8/1802 | 14/1849 | 0.59 (0.25; 1.39) | |
| ≥35 | 2/479 | 9/431 | 0.20 (0.04; 0.93) | |
| BMI (kg/m²) | | | | 0.25 |
| <30 | 6/1906 | 18/1852 | 0.32 (0.13; 0.81) | |
| ≥30 | 3/246 | 4/301 | 0.91 (0.21; 4.02) | |
| Fetal sex | | | | 0.10 |
| Boys | 5/1228 | 18/1194 | 0.27 (0.10; 0.72) | |
| Girls | 5/1053 | 5/1086 | 1.03 (0.30; 3.56) | |

Favours induction of labour    Favours expectant management

### B. Caesarean delivery

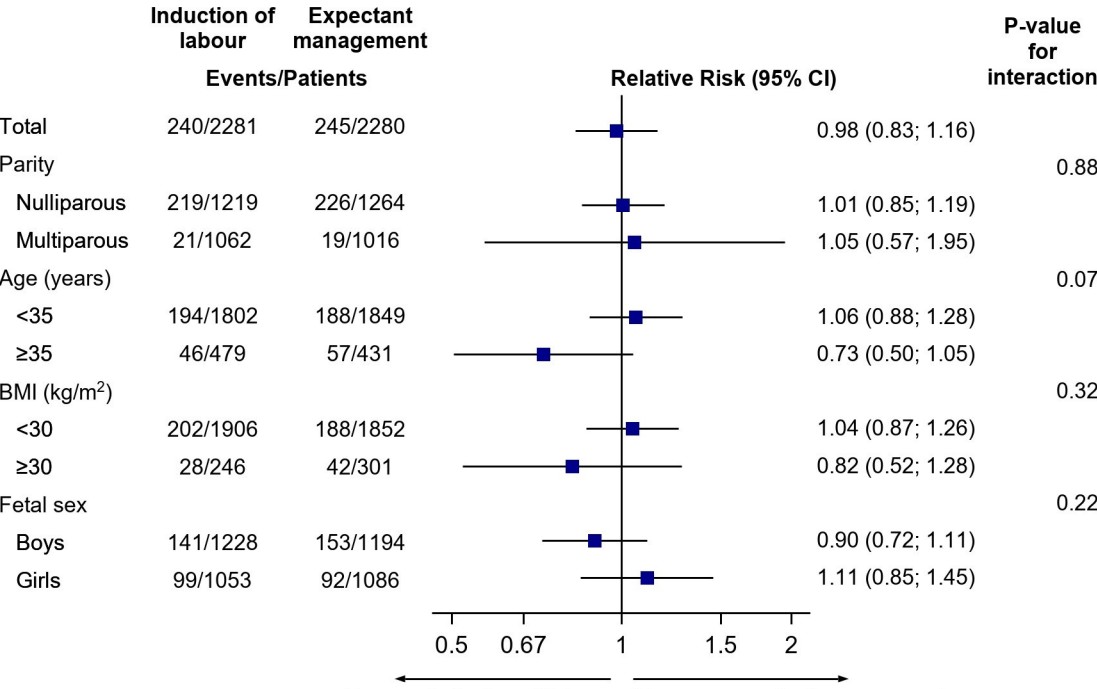

| | Induction of labour | Expectant management | Relative Risk (95% CI) | P-value for interaction |
|---|---|---|---|---|
| | Events/Patients | | | |
| Total | 240/2281 | 245/2280 | 0.98 (0.83; 1.16) | |
| Parity | | | | 0.88 |
| Nulliparous | 219/1219 | 226/1264 | 1.01 (0.85; 1.19) | |
| Multiparous | 21/1062 | 19/1016 | 1.05 (0.57; 1.95) | |
| Age (years) | | | | 0.07 |
| <35 | 194/1802 | 188/1849 | 1.06 (0.88; 1.28) | |
| ≥35 | 46/479 | 57/431 | 0.73 (0.50; 1.05) | |
| BMI (kg/m²) | | | | 0.32 |
| <30 | 202/1906 | 188/1852 | 1.04 (0.87; 1.26) | |
| ≥30 | 28/246 | 42/301 | 0.82 (0.52; 1.28) | |
| Fetal sex | | | | 0.22 |
| Boys | 141/1228 | 153/1194 | 0.90 (0.72; 1.11) | |
| Girls | 99/1053 | 92/1086 | 1.11 (0.85; 1.45) | |

Favours induction of labour    Favours expectant management

**Fig 2. Prespecified and post hoc subgroup analysis on (A) primary outcome: severe adverse perinatal outcome and (B) cesarean delivery.** *Including perinatal mortality, Apgar<4 at five minutes, HIE II-III, intracranial haemorrhage, neonatal convulsions, MAS, obstetric brachial plexus injury, and mechanical ventilation within 72 hours. BMI, body mass index; CI, confidence interval; HIE, hypoxic ischemic encephalopathy; MAS, meconium aspiration syndrome.

management until 42 weeks from 1.0% to 0.4%. The NNT to avoid one of these events is 175 (95% CI 94 to 1,267). The risk of perinatal death was reduced from 0.4% to 0.04% after IOL at 41 weeks with an NNT of 326 (95% CI 177 to 2,014). Also, the risk for an infant to be treated in neonatal care for four or more days was lower in the IOL group (NNT 103 [95% CI 59 to 385]). Cesarean delivery or operative vaginal delivery rates were comparable, just as most important maternal adverse outcomes. The low rate of cesarean delivery reflects the low-risk population of this IPD-MA. The rate of hypertensive disorders of pregnancy was found to be lower in the IOL group with a NNT of 57 (95% CI 39 to 106).

In the subgroup analysis, we found that nulliparous women had a significantly reduced risk of severe adverse perinatal outcome after IOL at 41 weeks compared with expectant management until 42 weeks. The results indicate that infants of low-risk nulliparous women reaching 41 weeks of pregnancy will probably benefit from IOL at 41 weeks. For infants of low-risk multiparous women, the incidence of a severe adverse perinatal outcome is low. It is not clear if they would benefit from IOL or not. In this IPD-MA, we lack the power to detect a difference in multiparous women regarding severe adverse perinatal outcome, but also with an adequate sample size, the NNT or to harm would probably be high. An explanation of the lower incidence of adverse outcomes in multiparous women might be that women with previous cesarean delivery were excluded from this IPD-MA. Furthermore, multiparous pregnant women at 41 weeks are probably women without major risk factors following from a previous pregnancy. However, there may also be a true difference in risk between nulliparous and multiparous women. The fact that infants of nulliparous women are at increased risk of adverse outcomes in late-term pregnancy is in agreement with some but not all previous studies [28–30].

In the post hoc subgroup analysis on fetal sex, we found that the boys, rather than girls, might benefit from IOL regarding the composite severe adverse perinatal outcome. However, perinatal mortality did not differ by fetal sex; it occurred in five boys and four girls in our IPD-MA.

The overall better outcome for IOL in our IPD-MA is in line with the latest Cochrane review on IOL at or beyond term; however, no difference between parity was found [9]. The increased risk of stillbirth with advancing gestational age is also shown in a recent large MA of cohort studies by Muglu and colleagues [1]. Several large observational studies are also in line with our findings [31,32]. However, other observational studies are not [33,34]. The Cochrane reviews included RCTs with different time frames of comparison, and most RCTs had upper limits of expectant management that went far beyond 42 weeks, which could at least partly explain the higher risk of perinatal mortality with a policy of expectant management [8,9].

## Strengths and limitations

IPD-MA is believed to be the preferred design for a systematic review of trials [35]. Our IPD-MA included two recently published large RCTs carried out in two high-income countries with a study population reflecting the general population in each country. Both RCTs excluded complicated pregnancies. Furthermore, the two countries, according to the Organisation for Economic Co-operation and Development, are comparable in life expectancy, level of education, perinatal mortality, and cesarean delivery. The overall similarity between the RCTs enabled us to redefine a primary composite outcome that more accurately reflects seriously affected neonates. We changed from Apgar <7 to Apgar <4 at five minutes because it is associated with an increased risk of long-term adverse neonatal outcome and replaced NICU admission by clearly diagnosed severe neonatal illness and severe complications because NICU admission criteria in Sweden and the Netherlands are not comparable [21]. In addition, the results of this IPD-MA are likely to be applicable to high-income countries due to the

equivalence in healthcare status in the included trials but less applicable to low- or middle-income countries. The heterogeneity between trials for most outcomes was low. Only for one important outcome (Apgar score) heterogeneity was considerable.

Risk of bias assessed by the authors themselves may be considered as not independent; however, the low risk of bias is in agreement with the assessment of the independent reviewers in the Cochrane 2020 updated review of Middleton and colleagues. [9].

There are a few issues, both in the SWEPIS trial and in the INDEX trial, which we would like to highlight. In both trials, only low-risk women were included, and women with former cesarean delivery or other major uterine surgery were excluded. Thus, how to advise these women is still unclear. The SWEPIS trial was stopped early due to safety reasons; therefore, the magnitude of the risk on perinatal death could be affected and may be overestimated also considering the difference in perinatal mortality rate between the SWEPIS and INDEX trial. Additionally, there was a discrepancy between the enrolment procedure of eligible low-risk women in different centres. In the Stockholm region, inclusion was performed after a routine ultrasonographic assessment, including measurement of fetal abdominal diameter and amniotic fluid, but at other centres, this was not performed routinely because it was not mandatory in the study protocol. One could argue that the women included in the Stockholm region (41% of all inclusions) were selected excluding women with fetuses at increased risk of adverse outcome. Perinatal mortality did not occur in the expectant management group in Stockholm centres (0/557; 0.0%), whereas in the other centres, there were six cases (6/822; 0.7%). However, there is currently insufficient evidence that routine surveillance with ultrasonographic assessment in late term in order to detect fetuses at risk reduces perinatal mortality [36–39]. In a Swedish retrospective study, a reduction in SGA but no reduction in rates of composite perinatal mortality and morbidity or stillbirth was found with routine ultrasound at 41 weeks compared with indicated ultrasound [38]. Thus, whether the use of routine ultrasonographic assessment at 41 weeks affected the outcome is difficult to determine. Furthermore, it reflects the real clinical situation in Sweden regarding the management of prolonged pregnancy and therefore probably increases the generalisability of the SWEPIS trial to the Swedish population.

In both the SWEPIS and INDEX trials, fetal surveillance was performed according to local protocol between 41 and 42 weeks. This could also be considered as a strength, rather than a limitation, because it increases external validity of these pragmatic trials. In the SWEPIS trial, it usually included an antenatal visit performed by a midwife with a clinical assessment and auscultation of fetal heart rate. In the INDEX trial, it generally included clinical assessment and assessment of the fetus with ultrasound, cardiotocography, and extra checks between 41 and 42 weeks [40]. This could be reflected in the higher rate of medical inductions for fetal and maternal reasons between 41 and 42 weeks in the INDEX expectant management group compared with the expectant group of the SWEPIS trial, though whether this contributed to a lower perinatal mortality rate is unknown and makes us aware that further studies are needed to evaluate the effect of screening regimes in late-term pregnancy.

In the INDEX trial, randomisation was not stratified by parity, which resulted in a small difference in distribution of nulliparous women between the groups (slightly higher rate in the expectant management group), though after adjustment for parity similar, results were shown. In addition, due to the healthcare system, a discrepancy between the IOL and expectant management group regarding level of care was present. All induced women were treated in an obstetrician-led secondary care setting, while a large proportion of the women in the expectant management group started their delivery in midwifery-led primary care including home births. This might be considered as performance bias, but several studies have shown that the level of care does not seem to influence the delivery outcome. Women at risk of adverse

perinatal or maternal outcome are referred to secondary care (hospital); this risk selection is on the basis of the Dutch obstetric care system [41–44]. Also, all babies in the expectant management group with adverse outcome were born in secondary care after referral before or during delivery.

Furthermore, for our combined IPD-MA, a limitation is the low number of included women (*n* = 5,161) compared with the recent Cochrane review (*n* = 12,479). However, none of the trials except the Gelisen and the INDEX trials in the Cochrane review fulfilled our inclusion criteria for gestational age. Including IPD from trials with induction before or after 41 weeks and an expectant management policy beyond 42 weeks would have caused methodological problems such as selection bias.

Finally, due to the few RCTs eligible for this IPD-MA, the possibility of assessing severe adverse perinatal outcomes with few events and subgroup analysis was limited.

## Conclusions and implications

In this study, we found that, overall, IOL at 41 weeks improved perinatal outcome compared with expectant management until 42 weeks without increasing the cesarean delivery rate. This benefit is shown only in nulliparous women, whereas for multiparous women, the incidence of mortality and morbidity was too low to demonstrate any effect. The magnitude of risk reduction of perinatal mortality remains uncertain. Women with pregnancies approaching 41 weeks should be informed on the risk differences according to parity so that they are able to make an informed choice for IOL at 41 weeks or expectant management until 42 weeks.

## Supporting information

**S1 Fig. Aggregate meta-analysis of studies comparing induction of labour with expectant management regarding perinatal mortality.**
(PDF)

**S2 Fig. Aggregate meta-analysis of studies comparing induction of labour with expectant management regarding cesarean delivery.**
(PDF)

**S1 Table. Baseline characteristics per trial in the populations included in the individual participant data meta-analysis.**
(PDF)

**S2 Table. Perinatal outcome per trial in the populations included in the individual participant data meta-analysis.**
(PDF)

**S3 Table. Delivery outcome per trial in the populations included in the individual participant data meta-analysis.**
(PDF)

**S4 Table. Maternal outcome per trial in the populations included in the individual participant data meta-analysis.**
(PDF)

**S5 Table. Prespecified and post hoc subgroup analysis on primary outcome: severe adverse perinatal outcome, perinatal mortality, and cesarean delivery in the population included in the individual participant data meta-analysis by subgroup.**
(PDF)

**S6 Table. Data availability contact information.**
(PDF)

**S1 Text. PRISMA-IPD checklist of items to include when reporting a systematic review and meta-analysis of individual participant data (IPD).**
(PDF)

**S2 Text. PICO, study selection, search strategies, and references of excluded studies.**
(PDF)

## Acknowledgments

Therese Svanberg, HTA librarian at the Medical Library at Sahlgrenska University Hospital Gothenburg, Sweden, performed the literature search. Annika Strandell, MD, PhD, associate professor, Region Västra Götaland, HTA-centrum, Gothenburg, Sweden, provided the aggregated MA on perinatal mortality and cesarean delivery. Anders Elfvin, MD, PhD, associate professor, Department of Pediatrics, The Queen Silvia Children's Hospital, Sahlgrenska University Hospital, Gothenburg, Sweden, and Anton van Kaam, MD, PhD, professor, Department of Neonatology, Emma Children's hospital, Amsterdam UMC-AMC, University of Amsterdam, the Netherlands, advised on matters within the field of neonatology. All included women in both trials are to thank for making this IPD-MA possible.

## Author Contributions

**Conceptualization:** Mårten Alkmark, Judit K. J. Keulen, Joep C. Kortekaas, Christina Bergh, Jeroen van Dillen, Henrik Hagberg, Ben Willem Mol, Joris A. M. van der Post, Ulla-Britt Wennerholm, Esteriek de Miranda.

**Data curation:** Mårten Alkmark, Judit K. J. Keulen, Joep C. Kortekaas, Ruben G. Duijnhoven, Mattias Molin, Ulla-Britt Wennerholm, Esteriek de Miranda.

**Formal analysis:** Mårten Alkmark, Judit K. J. Keulen, Joep C. Kortekaas, Ruben G. Duijnhoven, Henrik Hagberg, Mattias Molin, Ulla-Britt Wennerholm, Esteriek de Miranda.

**Funding acquisition:** Henrik Hagberg, Ulla-Britt Wennerholm.

**Investigation:** Mårten Alkmark, Judit K. J. Keulen, Joep C. Kortekaas, Ulla-Britt Wennerholm, Esteriek de Miranda.

**Methodology:** Mårten Alkmark, Judit K. J. Keulen, Joep C. Kortekaas, Christina Bergh, Ruben G. Duijnhoven, Henrik Hagberg, Mattias Molin, Ulla-Britt Wennerholm, Esteriek de Miranda.

**Project administration:** Mårten Alkmark, Judit K. J. Keulen.

**Software:** Ruben G. Duijnhoven.

**Supervision:** Ulla-Britt Wennerholm, Esteriek de Miranda.

**Validation:** Mårten Alkmark, Judit K. J. Keulen, Joep C. Kortekaas, Christina Bergh, Jeroen van Dillen, Ruben G. Duijnhoven, Henrik Hagberg, Joris A. M. van der Post, Ulla-Britt Wennerholm, Esteriek de Miranda.

**Visualization:** Mårten Alkmark, Judit K. J. Keulen, Joep C. Kortekaas, Mattias Molin, Ulla-Britt Wennerholm, Esteriek de Miranda.

**Writing – original draft:** Mårten Alkmark, Judit K. J. Keulen.

**Writing – review & editing:** Mårten Alkmark, Judit K. J. Keulen, Joep C. Kortekaas, Christina Bergh, Jeroen van Dillen, Ruben G. Duijnhoven, Henrik Hagberg, Ben Willem Mol, Joris A. M. van der Post, Sissel Saltvedt, Anna-Karin Wikström, Ulla-Britt Wennerholm, Esteriek de Miranda.

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
