## [Editor Report · Decision Letter 0]

22 Jul 2020

Dear Dr Alkmark, 

Thank you for submitting your manuscript entitled "Induction of labour at 41 weeks or expectant management until 42 weeks; an individual participant data meta-analysis of randomised trials" for consideration by PLOS Medicine.

Your manuscript has now been evaluated by the PLOS Medicine editorial staff and I am writing to let you know that we would like to send your submission out for external assessment.

Kind regards,

Richard Turner, PhD

Senior editor, PLOS Medicine

rturner@plos.org

---

## [Decision Letter · Decision Letter 1]

14 Aug 2020

Dear Dr. Alkmark,

Thank you very much for submitting your manuscript "Induction of labour at 41 weeks or expectant management until 42 weeks; an individual participant data meta-analysis of randomised trials" (PMEDICINE-D-20-03510R1) for consideration at PLOS Medicine. 

Your paper was evaluated by the editors and sent to independent reviewers, including a statistical reviewer. The reviews are appended at the bottom of this email and any accompanying reviewer attachments can be seen via the link below:

[LINK]

In light of these reviews, we will not be able to accept the manuscript for publication in the journal in its current form, but we would like to invite you to submit a revised version that addresses the reviewers' and editors' comments fully. You will appreciate that we cannot make a decision about publication until we have seen the revised manuscript and your response, and we expect to seek re-review by one or more of the reviewers. 

We hope to receive your revised manuscript by Sep 04 2020 11:59PM. Please email us (plosmedicine@plos.org) if you have any questions or concerns.

Please let me know if you have any questions. Otherwise, we look forward to receiving your revised manuscript in due course. 

Sincerely,

Richard Turner, PhD

rturner@plos.org

Please note PLOS' data policy (https://journals.plos.org/plosmedicine/s/data-availability) and add a non-author contact for readers interested, in the event of publication, in inquiring about access to study data. 

In the title, please replace the semicolon with a colon. 

In the abstract, please quote aggregate demographic details for study participants. 

Please add a new final sentence to the "methods and findings" subsection of your abstract, quoting 2-3 of the study's main limitations. 

At line 83, please begin the subsection with "In this study, we found that ..." or similar (and again at line 675). 

Where available, please quote p values alongside 95% CI throughout the paper. 

Rather than "p<0.0001", please quote exact p values or p<0.001 throughout the ms. 

Throughout the ms, please adapt reference call-outs to the following format: "... 16% [4,5]." (i.e., no spaces within the square brackets). 

Please use the spelling "fetus" (and derivatives) throughout the article.

Please abbreviate journal names in your reference list (e.g., "PLoS Med."). 

Please add a completed checklist for the most appropriate reporting guideline, which we imagine will be PRISMA, as a supplementary file (referred to in the methods section; "See S1_PRISMA_Checklist"). In the checklist, individual items should be referred to by section (e.g., "Methods") and paragraph number rather than by line or page numbers, as the latter generally change in the event of publication. 

Comments from the reviewers:

*** Reviewer #1: 

[See attachment]

Michael Dewey

*** Reviewer #2: 

Review of manuscript D-20-03510R1 for PLOS Medicine

Thank you for the opportunity to review the manuscript "Induction of labour at 41 weeks or expecting management until 42 weeks; an individual participant data meta-analysis of randomized trials", currently under consideration for publication in PLOS Medicine.

The aim of the study was to evaluate the effect of IOL at 41 weeks versus expectant management until 42 weeks on important and especially rare perinatal and maternal outcomes. We also aimed to assess whether treatment effects differed in subgroups.

Overall, the paper examine an important and complex topic and the paper is well-written. However, there are some major concerns when it comes to the methodology, data analysis, interpretation of the results, and the scientific ambitions of the paper.

I have elaborated my main concerns below:

Introduction:

In the introduction the body of evidence on the current topic is expected to be introduced for the reader. In the present introduction the two studies from the authors and the Cochrane metaanalysis are the overall main evidence. However, there are other randomized studies, meta-analysis (41 vs 42 weeks), and quasi-experimental studies but they are not presented and discussed - this is a major limitation of the study and need to be improved making a balanced introduction on the body of evidence to the reader.

In the last part of the introduction the aim of the study is presented - the authors want to study "important" and rare outcomes - but there are important outcomes not included in the analysis provided by the authors - I suggest the authors are more objective in their aim and reflect on which words they use.

Method:

In general, information on the limitations on the two studies included in the analysis are limited. This is a limitation as it have impact on how to treat data and make conclusions. Especially, the Swedish trial have several limitations (stopped early, violation of protocol etc) The authors do know of these limitations (also stated in the many responses to the published study) and should ensure transparency to these limitations in the present paper. A meta-analysis is not getting better that the study included in the analysis.

Outcome:

This study mainly use data from two previously published RCTs - one study from the Netherlands and a ¼ of a RCT study from Sweden. The main outcome is "rare perinatal and maternal outcomes" even though rare outcomes are very hard to examine in RCT (that's why we normally use other and more robust methods when studying mortality/rare outcomes) 

Rare outcomes has a nature of a fluctuate pattern in its distribution and therefor the possibility of stopping the trial earlier than expected is high and it follow with an overestimation of the studied effect - the authors acknowledge the risk of overestimation from the early stopped trial but do not take it into considerations when forming their conclusion -this is a major shortcoming of this present study. We do also know that the mortality in the swepis trial was higher than the mortality is in Sweden in general - pointing towards an overestimation of the effect and a fluctuate pattern. 

Another important issued with the outcome is when studying mortality we need to know more of the cases before we can calculate a number needed to treat (as the authors do) in other words if we conclude we are able to prevent some cases we need the case fatality not the total mortality as presented by the authors. I have had a look into stillbirth and perinatal deaths from week 41 and onwards in my country and found that some of the cases where not preventable - more information are therefor needed before number needed to treat is calculated and the conclusion needs to be moderated if we are not dealing with the case fatality (this is common knowledge in public health and mortality research but not always in the obstetric field)

Before we are able to make a meta-analysis it is important to access if the outcome cases fulfill the inclusion criteria - this is questionable for the Swedish trial as the UL assessment at 41+0 was not conducted for many of the cases. The authors needs to provide information on how many of the cases that fulfilled the inclusion criteria (UL assessment) at 41+0. There is some very small fetuses (SGA) which in normal practice would have been found if they have had an UL scan in 41+0 - meaning risk pregnancies were included in the trial (SGA children have 5-6 times higher mortality) violating the inclusion criteria.

Exposure:

What is the exposure in expectant management regime - this has not been specified - as this is important as it is the exposure - it is not enough to make a general statement as provided in the present manuscript - if the authors do not have a clear definition (and compliance) of the regime of expectant management in the two studies this needs to be clearly stated (see line 197) and conclusion may be downgraded if a clear statement of expectant management is not possible to describe. Description of expectant management may include different kinds of surveillance, number of antenatal visits etc)

Risk of bias:

In line 240 it is stated that risk of bias was assessed independently by some of the authors - the results of these scores are found in table 3 - both the Swedish trial and the trial from The Netherland is almost scores with low risk of bias in all categories despite of serious limitations especially in the Swedish trial affecting the level of bias in more than one category - more transparency is needed on the scoring process and maybe a more independently scoring process is needed to gain a more accurate and valid result.

Please also remember the low participation rate in both studies and how it affect the validity of the study 

Results:

It is well known that using composite outcome measures for rare outcomes in RCT is often inadequate in terms of extrapolation to clinical setting (but sufficient in relation to statistics one could say greater precision but with greater uncertainty) and the results gained more often apply to an individual measure rather to the composite. A high proportion of trials using composite outcome measures do not report significant results for the mortality component but for the composite outcome measure and further, very few of the trials with composite outcomes finds significant results for the mortality component but not for the composite outcome measure - this illustrate why composite outcome measure are very difficult to use when making clinical conclusions. The authors do not discuss this well known challenge and do neither take it into account when forming their conclusion. More information on this can be found in the publication by Freemantle or Cordoba or McKenna among others if the authors are not familiar with this challenge. 

In line 435, AAR would be a better measure to calculate - in this case it would be 0.6% - please include this number to gain transparency to the effect found.

In table 4, why is the Apgar <4 at 5 minutes (significant) used as a subcomponent in the primary composite outcome measure - why not Apgar < 7 at 5 minuts (not significant) - what is the result if Apgar <7 is used instead of Apgar < 4. Conventionally we use Apgar less than 7. Please also provide the distribution in the two study groups for Apgar < 4 and 7

In table 4, it is also illustrated that most of the death children are from the swepis trial meaning that it is mainly the swepis study (or a ¼ of a study with all it´s limitations) providing the main result for this study. As mentioned before there were some violation to the protocol meaning that most of the women did not have an ultrasound examination upon inclusion in 41+0 and detection of very small fetus was lacking meaning that high risk pregnancies were allocated to expecting management even though the inclusion criteria was low risk singleton pregnancies (we did experience the same in the term breech trial which have had devastating influence on women health and health care professionals expertise)

Table 4, the variable NICU > or equal to 4 days need further explanation - why the cutpoint of 4 days? Do the authors have indication for being admitted to NICU 

Table 5. there is only 3 days difference between induction of labour and expecting management - this need to be highlighted - as this may tell us that there is minor difference in gestational length in the two groups.

Table 5, I find a very low c section rate of 10% in both groups - this need to be mention together with the statement of no difference in c-section - as the low rate may leave no potential to study this outcome effectively. 

Table 5, the most common side-effect of the intervention studied (induction of labour) is hyperstimulation (proxy variable use of medication to slow down contractions) - this clinical relevant information is not included - do the authors have some information on hyperstimulation - if not this need to be stated as a limitation as it is a common side-effect of the intervention studied.

Table 5, meconium stained amniotic fluid (MSAF), 380 cases versus 52 cases - guess there is something wrong with the numbers - is 380 meant to be 38? - this outcome is significant - and this is expected as we will expect more MSAF with longer pregnancy - a physiological explanation - the outcome we worry about is aspiration of MSAF - please balance this (also in your conclusion) 

Figure 4, interaction is found for parity - this is quite normal. When interaction is found for parity all results from this study needs to be calculated separate for nulliparous and multiparous women as the effect is dependent by parity. The total calculation is biased and is not valid to form conclusions from.

Overall, the authors do not have a high internal validity of their study and therefor they need to discuss the limitations of each study and the merged study results better and how the limitations may affect the conclusion. At the moment the conclusion is not supported by the data provided in this paper. A clinical conclusion based on the data from the two studies may be vague as there is several limitations which do not allow for a clear conclusion neither recommendation for clinical practice.

In the present form, the paper is more a position statement paper rather than a scientific paper

There is a remarkable lack of balanced and relevant discussion of the findings - this is a shame as I expect the authors have had a huge work with data cleaning. 

*** Reviewer #3: 

This is a well-conducted and well-written piece of work on a topic of considerable importance, namely at what gestation is induction of labour (one of the most common obstetric procedures) in a post-dates pregnancy of demonstrable benefit regarding major adverse perinatal outcomes, and are there maternal adverse outcomes either caused or avoided with IOL? The authors have applied the gold-standard of IPD-MA to attempt to answer this question, and have used this to advantage in pre-specifying a composite adverse perinatal outcome that truly reflects substantial morbidity, unlike the "softer" adverse outcomes such as Apgar at 5 minutes <7 that are often used in individual RCTs. The authors show evidence that, at least for nulliparous women, IOL at 41 weeks leads to better perinatal outcome without increases in obstetric procedures or adverse maternal outcomes (and in the case of hypertensive disorders of pregnancy, a decrease in the IOL group) - and with numbers needed to induce. These findings are of immediate applicability to counselling women regarding IOL in high-income countries/those with comparable maternity care systems.

I do however suggest minor revisions as detailed below:

1) Abstract, methods and findings: "Perinatal deaths occurred in one (0.0%)..." - suggest change this to "Perinatal deaths occurred in one (<0.1%)....". I realise that with the numbers involved the rounding down makes 0%, however given that there was in fact one perinatal death, it is less confusing to the reader to say <0.1% rather than 0.0%.

2) Author summary (why was this study done?). The first point notes that in observational studies IOL has been associated with increased risks. However, IOL RCTs for a variety of reasons in a variety of settings do not usually suggest these risks. Therefore suggest to balance the first point by "In observational studies, although not usually in interventional studies, IOL has been associated with....(etc)." 

In point 3 "According to a recent meta-analysis..." suggest minor change for clarity "According to a recent meta-analysis on aggregate data from randomised controlled trials (RCTs), perinatal mortality was lower after induction of labour at or beyond term compared with a policy of expectant management. However, the upper limit of gestational age for expectant management was not taken into account in this meta-analysis".

3) Author summary (what do these findings mean?). The second point needs to be better expressed, as surely women already have the possibility to choose the timing of their post-dates IOL? Maybe "Women can then make an informed choice regarding IOL at 41 weeks versus awaiting spontaneous onset of labour until 42 weeks." 

4) Introduction, first sentence (line 132/133) suggest minor wording change "When to induce labour in overdue or post-term pregnancy has been under debate for many years".

5) Introduction, line 146 - minor typo in IOL (the L is not capitalised).

6) Introduction, line 147 - replace "Yet" with "However".

7) Introduction, line 156-159 when talking about the recent Cochrane review update, needs a little rewording for clarity. I think what is being stated here is that you had already done your IPD-MA analysis and were writing up results when the Cochrane update was published? If so, suggest (or similar) "After results of the current study were already finalised, an update of this Cochrane review was published in which the main conclusion was similar to the previous version (9). Although subgroup analysis was performed for parity, this was not done for the 41-42 weeks' comparison in the Cochrane review".

8) Introduction line 162 and throughout - fetal and fetus not "foetal" or "foetus". Both in American English and UK English fetus has no "o". 

9) Methods, lines 413, 415-16 and 421 - I am not sure what is meant by "directness". I suspect from the context it means external validity/applicability to settings outside that of the trial itself - however please clarify. 

10) Result, table 5 - ?p value missing for gestational age at delivery (days)

11) Discussion, lines 597-99: suggest minor rewording for clarity - "In the post hoc subgroup analysis on fetal sex we found that boys rather than girls might benefit from IOL regarding the composite severe adverse perinatal outcome. However, perinatal mortality did not differ by fetal sex, occurring in 5 boys and 4 girls in our IPD-MA."

12) Discussion line 626 "advise" rather than "advice".

13) Discussion line 658 - "shown" rather than "showed".

14) Discussion, strengths and limitations: The additional limitation of external validity should be touched upon. These data are likely to be highly applicable to high-income countries, particularly those with comparable maternity care systems to The Netherlands and Sweden (i.e. most of Europe, Australia/New Zealand, Canada). They are less likely to be applicable to low or middle-income countries.

15) Please double-check referencing. Some journal names are missing appropriate capitals e.g. reference 20 should be JAMA not Jama.

***

[LINK]

---

## [Editor Report · Decision Letter 2]

7 Sep 2020

Dear Dr. Alkmark,

Thank you very much for submitting your revised manuscript "Induction of labour at 41 weeks or expectant management until 42 weeks: a systematic review and an individual participant data meta-analysis of randomised trials" (PMEDICINE-D-20-03510R2) for consideration at PLOS Medicine. 

We understand that you wish to submit a further revision to address some minor issues in the paper, and are hereby inviting you to do so. 

Please let me know if you have any questions - we look forward to receiving your revised manuscript shortly.

Sincerely,

Richard Turner, PhD

rturner@plos.org

---

## [Decision Letter · Decision Letter 3]

6 Oct 2020

Dear Dr. Alkmark,

Thank you very much for re-submitting your manuscript "Induction of labour at 41 weeks or expectant management until 42 weeks: a systematic review and an individual participant data meta-analysis of randomised trials" (PMEDICINE-D-20-03510R3) for consideration at PLOS Medicine.

I have discussed the paper with editorial colleagues and our academic editor, and it was also seen again by 3 reviewers. I am pleased to tell you that, provided the remaining editorial and production issues are fully dealt with, we expect to be able to accept the paper for publication in the journal.

[LINK]

Please let me know if you have any questions. Otherwise, we look forward to receiving the revised manuscript shortly. 

Sincerely,

Richard Turner, PhD

rturner@plos.org

Requests from Editors:

Please incorporate the revised data statement from your point-by-point response into the resubmission form.

Early in the "Methods and findings" subsection of your abstract, please specify the date of the literature search. 

At the end of the abstract, please make that "Study registration". 

At line 90, please make that "improved" (perinatal outcome); please make the same change at line 728. 

In the reference list, please revisit reference 13, which appears to contain some typos. 

Please ensure that all author names are spelt out, e.g., in reference 18.

Comments from Reviewers:

*** Reviewer #1: 

The authors have addressed my points and clarified some of the things which were puzzling me.

I still think that it is illogical to carry out a statistical test for something which was fixed by design but I would not want to overplay this. Many things which seem odd to me are commonplace to others.

Michael Dewey

*** Reviewer #2: 

[see attachment]

*** Reviewer #3: 

All comments addressed, thank you.

***

[LINK]

---

## [Editor Report · Decision Letter 4]

26 Oct 2020

Dear MD Alkmark, 

On behalf of my colleagues and the academic editor, Dr. Persson, I am delighted to inform you that your manuscript entitled "Induction of labour at 41 weeks or expectant management until 42 weeks: a systematic review and an individual participant data meta-analysis of randomised trials" (PMEDICINE-D-20-03510R4) has been accepted for publication in PLOS Medicine. 

PRODUCTION PROCESS

Before publication you will see the copyedited word document (within 5 business days) and a PDF proof shortly after that. The copyeditor will be in touch shortly before sending you the copyedited Word document. We will make some revisions at copyediting stage to conform to our general style, and for clarification. When you receive this version you should check and revise it very carefully, including figures, tables, references, and supporting information, because corrections at the next stage (proofs) will be strictly limited to (1) errors in author names or affiliations, (2) errors of scientific fact that would cause misunderstandings to readers, and (3) printer's (introduced) errors. Please return the copyedited file within 2 business days in order to ensure timely delivery of the PDF proof. 

If you are likely to be away when either this document or the proof is sent, please ensure we have contact information of a second person, as we will need you to respond quickly at each point. Given the disruptions resulting from the ongoing COVID-19 pandemic, there may be delays in the production process. We apologise in advance for any inconvenience caused and will do our best to minimize impact as far as possible.

PRESS

PROFILE INFORMATION

Thank you again for submitting the manuscript to PLOS Medicine. We look forward to publishing it. 

Best wishes, 

Richard Turner, PhD

Senior Editor 

PLOS Medicine

plosmedicine.org